# Efficacy of Computed Tomography and Magnetic Resonance Imaging in the Assessment of Depth of Invasion in Oral Squamous Cell Carcinoma: Analysis of 125 Clinical Cases

**DOI:** 10.3390/diagnostics13233578

**Published:** 2023-12-01

**Authors:** Adam Michcik, Adam Polcyn, Łukasz Garbacewicz, Tomasz Wach, Maciej Sikora, Marta Bień, Barbara Drogoszewska

**Affiliations:** 1Department of Maxillofacial Surgery, Medical University of Gdansk, Mariana Smoluchowskiego 17, 80-214 Gdansk, Poland; adampolcyn@gumed.edu.pl (A.P.); lgarbacewicz@gumed.edu.pl (Ł.G.); bienmartaa@gmail.com (M.B.); drog@gumed.edu.pl (B.D.); 2Department of Maxillofacial Surgery, Medical University of Lodz, Zeromskiego 113, 90-549 Lodz, Poland; tomasz.wach@umed.lodz.pl; 3Department of Maxillofacial Surgery, Hospital of the Ministry of Interior, Wojska Polskiego 51, 25-375 Kielce, Poland; sikora-maciej@wp.pl

**Keywords:** oral squamous cell carcinoma, cancer imaging, CT, MRI, DOI, tumor size, nodal metastases

## Abstract

Radiological diagnosis of oral squamous cell carcinoma (OSCC) is one of the main steps in treatment planning. T (tumor size), DOI (depth of invasion) (AJCC 8th edition), and nodal metastases (N+) were evaluated using CT and MRI to assess the most effective imaging method. The effectiveness of the radiological imaging methods was compared with histopathological results. Imaging diagnostic studies were performed and retrospectively analyzed in 125 patients with OSCC (CT *n* = 54 and MRI *n* = 71). Histopathology evaluated T, DOI, and N+. The radiological T results of CT in comparison with histopathological examination showed agreement in 62.5% of cases for T1, 56.25% for T2, 25% for T3, and 42.8% for T4 (*p*-value = 0.07), and regarding MRI, 52.2% for T1, 36.4% for T2, 33.3% in T3, and 33.3% for T4. The DOI results of CT and MRI juxtaposed against the histopathological findings were as follows: for CT, *n* = 18 for DOI ≤ 10 mm and *n* = 36 for >10 mm; for MRI, *n* = 29 for DOI ≤ 10 mm and *n* = 42 for >10 mm (DOI CT vs. DOI hist. pat. *p*-value = 0.23; DOI MRI vs. DOI hist. pat. *p*-value = 0.006). Regarding nodal metastasis, *n* = 21 for N0 and *n* = 32 for N+ for CT (*p*-value = 0.02), and *n* = 49 for N0 and *n* = 22 for N+ for MRI (*p*-value = 0.1). In the radiological N+ group, the histopathological findings coincided with the results of MRI and CT in 27% and 62.5% of cases, respectively (N0: 83.6% for MRI; 85.7% for CT). Upon evaluating T, a decreasing percentage of overlapping results with an increasing tumor size was observed. The accuracy of both imaging studies was at a similar level, with a slight advantage for MRI. Among the patients on whom CT was performed, DOI analysis did not show statistically significant differences. This led to the conclusion that, in most cases, the DOI results based on CT overlapped with those described via histopathological examination. However, among the group of patients with MRI as the imaging method of choice, the differences proved to be statistically significant (*p*-value = 0.006). The results of this study indicate that CT is a more accurate method for DOI assessment. The results of the radiologic metastasis evaluation (N+ group) overlapped more in the CT group, while the percentage of corresponding results in the radiologic N0 vs. hist. pat. N0 group was high and similar in both groups. These results indicate that it is easier to confirm the absence of a metastasis than its presence.

## 1. Introduction

The treatment of OSCC (oral squamous cell carcinoma) remains a therapeutic challenge, and long-term results are not satisfactory [1]. Qualifying a patient with OSCC for surgical treatment is an extremely important part of the holistic therapeutic process. By analyzing a patient’s imaging studies, the treatment plan and the extent of resection of the tumor and neck lymphatic system are outlined. Moreover, the possibility of achieving oncologic radicality, which is a cardinal factor influencing 5-year survival, is evaluated. Thus, effective imaging studies are of extreme importance. In addition to evaluating the primary tumor, the radiological examination must allow for an accurate assessment of the neck lymphatic system. Approximately 40% of patients with OSCC at the time of diagnosis already present with metastatic lymph nodes [2], which are known to significantly reduce a patient’s prognosis [3]. In addition to evaluating the cervical lymphatic system, it is crucial to accurately assess the tumor’s features, especially the depth of invasion (DOI), which is a prognostically significant factor [4,5]. A classification published in the eighth edition of the American Joint Committee on Cancer’s (AJCC) manual included depth of invasion due to the strong association of the DOI with nodal metastasis [6].

Tumors in the facial part of the skull, including the oral cavity, become inoperable relatively quickly, and the surgical treatment of extensive tumors is associated with a decreased quality of life for patients [7,8]. For this reason, the meticulous performance of imaging studies in patients diagnosed with OSCC is essential and, in some cases, determines the initiation or withdrawal of surgical treatment. This is of the utmost importance, since abandoning surgical treatment significantly worsens a patient’s prognosis, and using radiotherapy as the only treatment modality reduces the 5-year survival rate [9].

In the case of tumors presenting with endophytic growth, surgeons frequently face the difficulty of qualifying a patient for surgery solely based on the results of imaging studies. Cases of extensive OSCC are particularly demanding when the possibility of radicality is determined based on imaging studies alone. A presurgical DOI assessment may include standard tests such as MRI or CT [10,11,12]. Intraoral ultrasound (US) is a useful and accurate method of DOI assessment [13], but its use is limited to locations allowing free access to the ultrasound head [14].

Similar to full-thickness biopsies [15], US has diagnostic value for small OSCC. However, in the case of extensive tumors, these methods do not demonstrate high DOI imaging efficiency.

This prompted the authors to conduct a detailed evaluation of the accuracy of magnetic resonance imaging (MRI) and computed tomography (CT) scans performed preoperatively in patients qualified for surgery and to compare these results with the histopathological findings of these patients.

## 2. Materials and Methods

The analysis included 125 patients with OSCC, diagnosed and qualified for surgical treatment at the Department of Maxillofacial Surgery of the Medical University of Gdansk in the years 2017–2019. In the studied group, *n* = 54 were evaluated for CT and *n* = 71 for MRI. The maximum time between radiological examination and surgical treatment was 3 weeks; however, most patients were operated on within two weeks, which minimized the risk of false results for having too long a period between examinations and surgical treatment. The standard protocol at our institution for patients with head and neck cancers includes CT or MRI (head and neck) with intravenous administration of contrast, chest CT, and abdominal ultrasound (Table 1).

### 2.1. Imaging

All patients underwent routinely performed presurgical contrast-enhanced CT or MRI of the head and neck region. No patients presented with generally known contraindications for MRI, CT, and intravenous contrast administration. The scanned region consisted of the maxilla-facial and neck regions to the level of the tracheal bifurcation.

CT exams were performed with a Siemens Somatom Definition Flash (Siemens Healthcare, Erlanger, Germany) using the standard contrast-enhanced protocol.

MRI was performed using a 1.5 T MR Magnetom Aera Flash (Siemens Healthcare, Germany) with the dedicated coil.

The sequences were performed per the standard head and neck examination protocol established in our institution with diffusion-weighted sequences (DWI) and T1-weighted DIXON images before and after intravenous contrast administration.

Measurements were independently performed by two maxillofacial surgeons with 15 years of experience (at two different maxillofacial surgery centers). The results were compared with the imaging data and, through mutual consensus, were determined and used for further analysis.

A classification published in the eighth edition of the American Joint Committee on Cancer’s (AJCC) manual includes the DOI in T. However, for this study, the DOI was assessed separately.

The DOI was measured (in T1-weighted DIXON images and T2-weighted fat-saturated images) from the level of the mucosal surface adjacent to the tumor to the deepest point of invasion in axial or coronal scans, depending on the location of the tumor. The histopathological DOI was measured from the basement membrane of the epithelium using an imaginary line connecting it to the basement membrane of normal squamous cells (Protocol for the Examination of Specimens from Patients with Cancers of the Oral Cavity; version 4.2.0.0). This study excluded deeply ulcerated tumors, where the radiological measurements are much smaller than those of histopathological and exophytic tumors, and the histopathological DOI is 0. This allowed for a reliable assessment of the effectiveness of radiological imaging in relation to histopathological results. Due to the criteria used, the concept of tumor thickness (TT), which, according to the Eighth Edition AJCC manual, is not included in the clinical classification, was deliberately not introduced into this study. The tumor size was assessed in two orthogonal axes according to the Eighth Edition AJCC classification. All tumors assessed in this study were soft tissues (tongue and floor of the mouth), and no bone infiltration was assessed (Figure 1).

Cervical lymph nodes were assessed for shape, margins, enlargement, the presence of necrosis, and the pathologic enhancement pattern. The obtained data were collected and compared with the histopathological results after surgery. The histopathological evaluation considered the T, DOI, and N+ confirmation of nodal metastasis to the lymphatic system of the neck.

### 2.2. Statistics

Statistical analysis was performed using the following tests: for qualitative data, the two-sided Fisher test and McNemar’s test were performed, whereas for quantitative data, two-sided Student’s *t*-test (two-sided *t*-test), the two-sided Mann–Whitney (also called the two-sided Wilcoxon rank-sum test) test, tests of variance, one-way ANOVA, and the Kruskal–Wallis rank-sum test were applied. This study’s premise was to evaluate how effective and accurate the two radiological imaging methods are and to what extent they coincide with postoperative histopathological findings, as well as whether their imaging accuracy changes with increasing values of tumor parameters (T and DOI).

## 3. Results

Multivariate analysis showed that the results of presurgical imaging in some of the analyzed cases did not coincide with the postsurgical histopathological results.

### 3.1. T—Tumor Size

This study revealed the presence of significant differences in the assessment of T depending on the type of examination used.

MRI

In the case of MRI (T1 *n* = 23, T2 *n* = 33, T3 *n* = 12, and T4 *n* = 3), there was an apparent tendency toward discrepancies in T assessments in imaging studies compared with T assessments presented in histopathological results. This incoherence increased with increasing tumor size (52.2% for T1, 36.4% for T2, 33.3% for T3, and 33.3% for T4) (Figure 2).

CT

Comparing the results of T from CT (T1 *n* = 8, T2 *n* = 16, T3 *n* = 16, and T4 *n* = 14) and histopathological examination, statistically significant differences were revealed (*p*-value = 0.07 for McNemar’s test; interpretation of G-Cohen score = 0.30) (Figure 2), as well as a decreasing accuracy of measurements with tumor growth (62.5% for T1, 56.25% for T2, 25% for T3, and 42.8% for T4). The exception was T4 tumors, which is attributed to the inclusion of patients with extensive tumors classified as T4.

This provides compelling evidence that both imaging methods most accurately depicted small T1 and T2 tumors.

### 3.2. DOI

MRI

In the next step of the analysis, the results of the DOI measurements in MRI (≤10 mm *n* = 29; >10 mm *n* = 42) and CT (≤10 mm *n* = 18; >10 mm *n* = 36) were compared with the DOI measurements obtained from the postoperative histopathological results. Patients were divided into two groups, DOI > 10 mm and ≤10 mm. Juxtaposing the radiological DOI measurements in MRI versus the histopathological results, discrepancies in the measurements were obtained, and the differences proved to be statistically significant in the results of the compared groups (*p*-value = 0.006) (Figure 3).

CT

The results in the group of patients who underwent CT scans were quite different. There were no statistically significant differences between the two study groups (DOI CT vs. DOI hist. pat. *p*-value = 0.23 for McNemar’s test) (Figure 3), and the obtained results in both groups were similar and overlapped in most cases. Most of the studies available in the literature indicate MRI as a method with higher accuracy in DOI evaluation. [16,17].

These results encourage a debate on the effectiveness of the radiological assessment of the DOI. The choice of imaging method depends on many factors. Naming an unequivocally superior method of examination that is of higher accuracy in all clinical cases is impossible. One must consider various factors when choosing the type of examination method, including the location and T, the condition of the patient’s dental health, and the presence of fixed prosthetic metal components in the oral cavity.

### 3.3. N

The subsequent stage of analysis included the lymphatic system of the neck. The evaluation of cervical lymph nodes was as follows: for MRI, N0 *n* = 49, N+ *n* = 22, and *p*-value = 0.1 for McNemar’s test; for CT, N0 *n* = 21, N+ *n* = 32, *p*-value = 0.02 for McNemar’s test, and interpretation of G-Cohen score = 0.30 (Figure 4).

The radiological assessment of the neck lymphatic system in imaging studies was characterized by a significantly higher percentage of overlapping findings in the N0 group, i.e., 83.6% for MRI N0 and 85.7% for CT N0. In the case of radiological N+, the confirmation of nodal metastasis via histopathology was 27% for MRI and 62.5% for CT (Table 2). It should be noted that patients with clustered and dissected lymph nodes, whose radiological and clinical evaluations did not raise doubts about the presence of metastasis, were disqualified from the analyzed group. This study’s premise was to investigate nodal metastases that were clinically non-palpable and without major local advancement in the neck.

## 4. Discussion

The performed analysis proved that the results of the radiological examination of OSCC patients were not always confirmed via postoperative histopathological investigation. Analyzing the radiological T in MRI, the number of results overlapping with those achieved in histopathological studies decreased as the T increased (52.2% for T1, 36.4% for T2, 33.3% for T3, and 33.3% for T4), which may reflect the greater difficulty of assessing the borders of large tumors in MRI. In contrast, Piia Huopainen et al. came to the opposite conclusion, studying a group of 200 patients in terms of the accuracy of assessing tumor size in MRI and noting the lowest correlation between results in T1 tumors [18]. According to our study, radiological evaluation of small OSCCs demonstrates a lower risk of measurement error and an easier capture of tumor boundaries. The available literature confirms that T1-weighted images before contrast administration are useful in differentiating small oral tumors from surrounding adipose tissue or bone marrow involvement [19] and should often be used for imaging small tumors. In our study, we used T1- and T2-weighted images and obtained the best results in the T1 tumor group. However, MRI can overestimate the size of a tumor due to hemorrhage or inflammatory changes [20], which may translate into lower accuracy in assessing extensive tumors.

Upon analyzing the radiological T results from the CT examinations, differences were noted regarding the histopathological results (*p*-value = 0.07). T1 tumors had concordant results in 62.5% of cases, T2 in 56.25%, T3 in 25%, and T4 in 42.9%. As with MRI, T1 tumors overlapped in the highest percentage in the studied group, which may reflect, as with MRI, the greater difficulty of defining the boundaries of large tumors. In addition, CT scanning risks greater measurement error due to artifacts generated by dental materials [21] and exposes the patient to a significant dose of X-rays [22]. Considering the above factors, in certain cases (multiple amalgam fillings, metal dental bridges, and titanium implants) CT examinations will be unreadable and subject to a high risk of measurement error. In such cases, we should refrain from performing a CT examination. As such, with the comparable results of the evaluations obtained using MRI vs. CT, MRI examination is a more favorable tool for the preoperative imaging of T. However, should the location of the OSCC be of great importance when choosing an OSCC imaging method?

The idea that tumors infiltrating bone were better imaged with CT was held for many years. However, as MRI increases in accuracy, this view is losing relevance, and numerous publications state the effectiveness of MRI in imaging bony structures [23,24]. Conversely, soft-tissue imaging using CT has also become more accurate as devices increase in resolution. Currently, this method is used not only in oncology but also in other branches of maxillofacial surgery, such as traumatology and the soft-tissue imaging of orbital fractures [25]. In the opinion of the authors and based on the above information, the choice of imaging method cannot only depend on the primary location of the tumor. The assessment of the DOI, which is an extremely prognostically important factor [4,26,27], and its presurgical evaluation is an important part of surgical treatment planning among patients with OSCC. Our study showed that the results of DOI assessment in MRI (MRI: ≤ 10 mm *n* = 29; hist. pat.: ≤ 10 mm *n* = 18; DOI > 10 mm: MRI *n* = 42, hist. pat. *n* = 14; *p*-value = 0.006 for McNemar’s test) proved to be statistically significant. As the DOI increased, the percentage of results overlapping (62% for ≤10 mm; 33.3% for >10 mm) with the DOI stated in the subsequent histopathological examination decreased. This demonstrates the increasing difficulty of capturing the tumor boundary in large tumors of the endophytic growth type, which translates into a risk of poor DOI estimation before planned surgical treatment. It should be noted, however, that histopathological DOI assessment according to the Eighth Edition AJCC guidelines is not easy in every case. In the case of an extensive tumor, the sectioned preparation of which does not contain the epithelium surrounding the tumor, determining an imaginary line connecting the basement membrane of normal squamous cells is impossible, and in the case of a polypoid tumor, the DOI will be zero [28]. This limits the ability to compare radiological and histopathological DOIs. The patient qualification criteria used in our study (patients with deeply ulcerated and exophytic tumors were excluded) allowed such an analysis to be performed.

The literature provides information on the results of DOI assessment in MRI. Lwin et al. found in their study that most of the tumors studied appeared thicker in MRI than in histopathological examinations [29]. This is of utmost importance, as the DOI is known to be strongly associated with the risk of occult nodal metastasis and nodal recurrence. Murakami et al. demonstrated in their study the absence of nodal recurrence in patients with a DOI of <5 mm in the evaluated group [30]. Moreover, the DOI score in imaging studies may be an indicator determining whether an elective neck dissection should be performed or abandoned [26].

According to the authors, elective neck dissection should be performed in each patient presenting with OSCC that enhances after contrast agent administration with a location in the tongue or floor of the mouth. According to some authors, the DOI score in MRI should be an indicator deeming nodal surgery to be necessary in cases with a DOI of >7.5 mm [31]. Balasubmaranian et al. indicate that elective nodal surgery is necessary for oral floor tumors with a DOI of >2 mm and tongue of >4 mm [32]. Furthermore, the probability of nodal recurrence is correlated with the DOI. Minamitake et al. determined that the 5-year probability of nodal recurrence in a group of patients with a DOI of ≤5 mm was 4%, while in the >5 mm group it was 32.1% [33]. In our study, comparing DOI measurements in CT with DOI measurements in histopathological studies showed no statistical significance (DOI CT ≤ 10 mm *n* = 18; DOI hist. pat. ≤ 10 mm *n* = 12; DOI CT > 10 mm *n* = 36; DOI hist. pat. > 10 mm *n* = 25; *p*-value = 0.23). The results in the ≤10 mm group overlapped in 66.6% of cases, while those in the >0 mm group overlapped in 69.4%. CT with contrast agent administration provided high efficiency in assessing the DOI. Nevertheless, the results of the analysis proved CT to be an examination method that allows a more accurate assessment of a tumor’s DOI compared with MRI. This experience is shared by other authors [34,35,36], affirming CT as a test with high diagnostic accuracy for DOI.

As this paper and those available in the literature indicate, DOI assessment is a key component of a patient’s overall treatment and an important factor in patients’ 5-year survival. The choice of DOI imaging modality should depend on several factors, including the location and size of the tumor, the patient’s dental status, the presence of fixed metal prosthetic components, and general contraindications to any of the tests. The presence of metastasis in the neck lymphatic system is a hallmark of OSCC, and determining the loco-regional spread of the tumor is of crucial value. The radiological assessment of the neck lymphatic system in imaging studies was characterized by much lower percentages of overlapping findings in the N+ group, i.e., 27% for MRI and 62.5% for CT. In the case of radiological N0, confirmation of the absence of nodal metastasis upon histopathological examination occurred correspondingly in MRI at 83.6% and CT at 85.7%. The higher percentages of overlapping results in the N0 group indicate that it is easier to exclude nodal metastasis than to confirm its presence.

These results show how difficult radiological evaluation of the neck lymphatic system can be, and metastatic lymph nodes do not have radiological features suggestive of metastasis at every stage. However, as Pakkanen et al. point out in their study, the radiological evaluation of small lymph nodes of <7 mm in diameter is extremely difficult [37]. The diagnosis of radiological nodal metastases in patients with early-stage disease remains a challenge. This is confirmed by the results of our study and others available in the literature [38]. In addition to CT and MRI, PET-CT is an examination method that can be used to depict the primary focus as well as the lymphatic system. Yang L. et al. stated in an evaluation of PET-CT in patients with small OSCC that the sensitivity and specificity of PET-CT in predicting metastasis in T1 tumors were 66.6 and 89.8% [39]. These results make PET-CT a highly effective imaging test, but its availability does not allow it to be performed in all patients with OSCC.

Histopathological findings of metastatic lymph nodes are an extremely important indicator of cancer progression, influencing 5-year survival rates [40]. According to the authors, despite the relatively high effectiveness of imaging studies, it is impossible to base a decision about the extent of surgery solely on radiological evaluation. Therefore, it can be concluded that the decision to not perform elective neck surgery cannot be based solely on the result of the radiological imaging of a patient’s lymphatic system. This is also confirmed by other studies available in the literature [29]. Some suggest performing both examinations (CT and MRI) to increase the imaging accuracy of the OSCC and the neck lymphatic system [41]. However, the authors of the present study do not share this view. The performance of both examinations prolongs the diagnostic period and exposes the patient to an additional dose of X-rays, and the decision on the scope of the surgical treatment is not made solely on the radiological evaluation. When analyzing the results of the tests performed, we cannot unequivocally state the superiority of one test over the other. Their effectiveness depends on many variables. As mentioned earlier, the choice of a radiological imaging method for a patient with OSCC should consider many other factors, i.e., the T and location of the primary tumor, the dental status of the patient, including the presence of fixed metal prosthetic components, and their general condition. Considering all the analyzed factors included in the presented study (T, DOI, and N+), we cannot unequivocally point to the one with higher efficacy and the choice should always be individual.

## 5. Conclusions

The present study emphasizes how difficult the preoperative radiological evaluation of OSCC can be despite the increasing accuracy of CT and MRI. Both studies showed relatively high imaging efficacy, but it is impossible to identify a study that is superior in all clinical cases. CT and MRI efficacy has been proven to decrease with tumor growth (T and DOI). This result should be considered when evaluating large tumors at the border of operability. Comparing the DOI assessment in both studies, the CT examination was slightly more accurate. The conducted analysis also showed that the radiological evaluation of the neck lymphatic system for the presence of nodal metastases is not always simple and does not always lead to the correct diagnosis. It has been shown that it is easier to diagnose the absence of a nodal metastasis (N0) than its presence (N+). Therefore, when evaluating the results, we cannot unequivocally state the superiority of one imaging method over the other, and the choice should be supported with a multivariate analysis carried out individually for each patient.

## Figures and Tables

**Figure 1 diagnostics-13-03578-f001:**
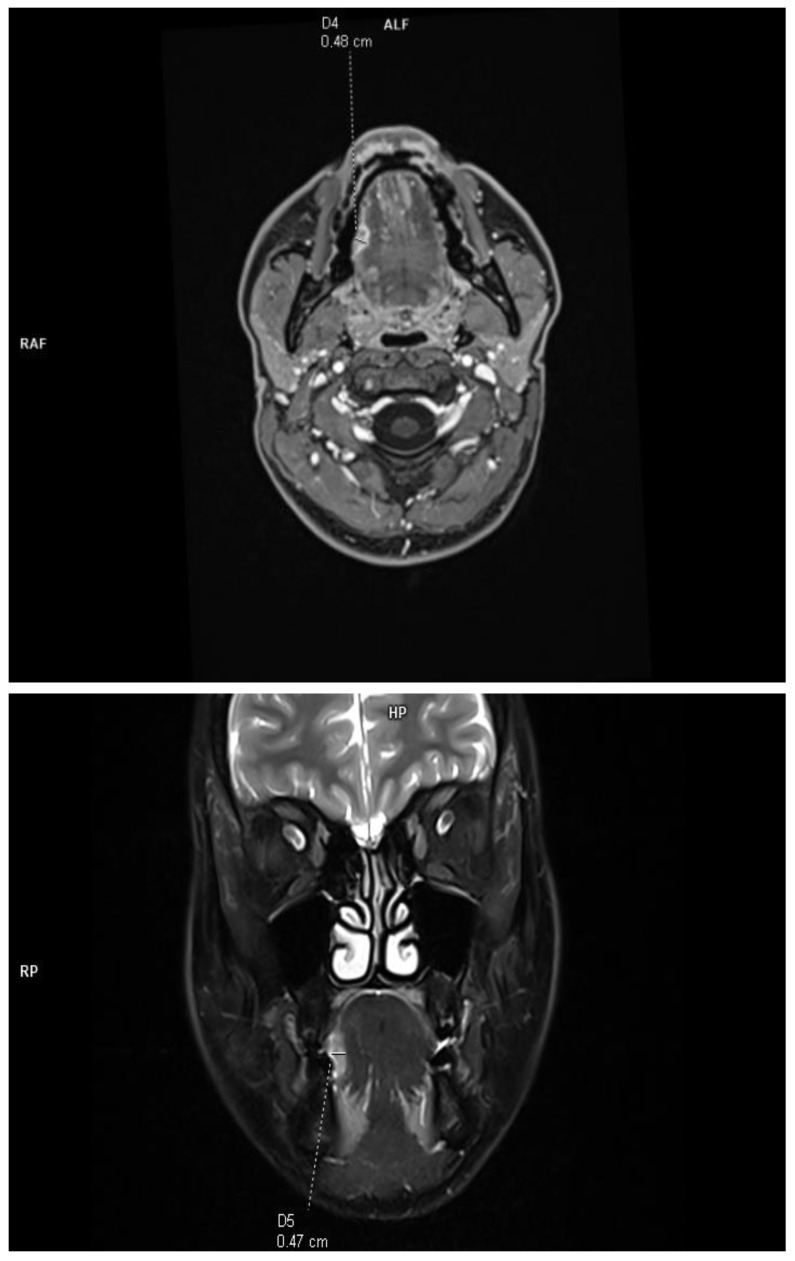
Example of DOI measurements performed in T1-weighted axial plane and T2-weighted coronal plane.

**Figure 2 diagnostics-13-03578-f002:**
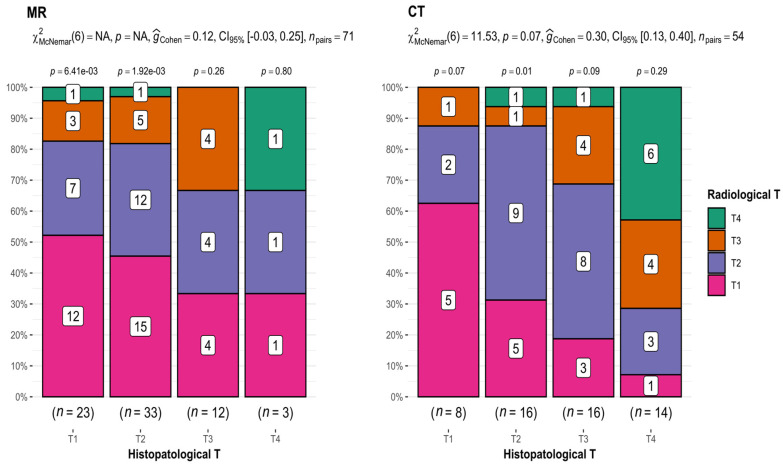
Comparison of radiological and histopathological T results. CT: *p*-value = 0.07; interpretation of G-Cohen score = 0.30.

**Figure 3 diagnostics-13-03578-f003:**
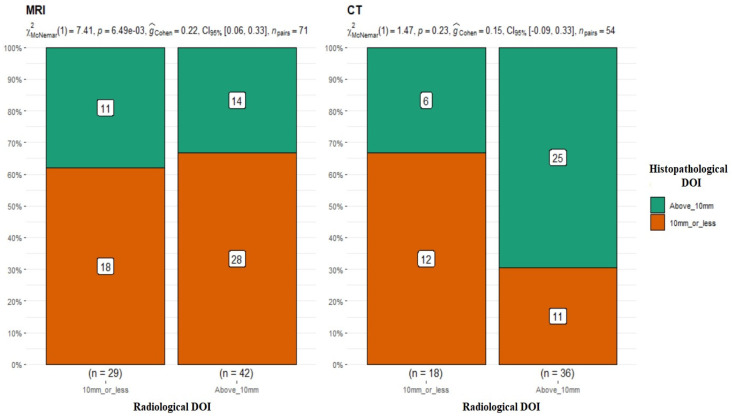
Comparison of radiological and histopathological DOI measurements. MRI: *p*-value = 0.006 for McNemar’s test; G-Cohen’s result = 0.22. CT: *p*-value = 0.23.

**Figure 4 diagnostics-13-03578-f004:**
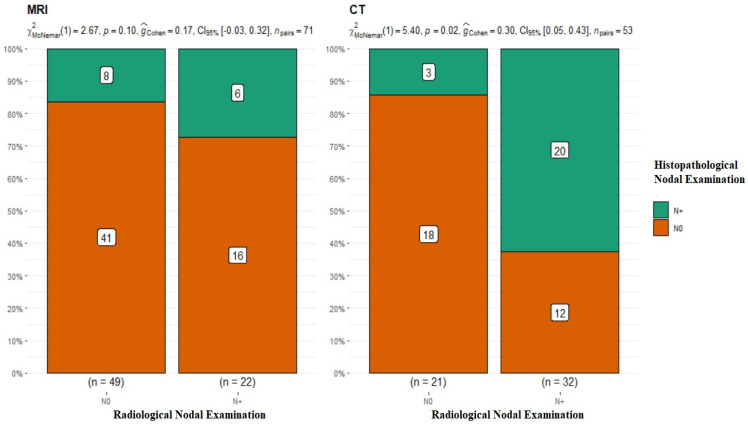
Summary of the radiological and histopathological results of the evaluation of the lymphatic system of the neck. MRI *p*-value = 0.1; CT *p*-value = 0.02.

**Table 1 diagnostics-13-03578-t001:** Characteristics of the study group.

Feature		*N*	%
Gender	Male	84	67.2
	Female	41	32.8
Age	50th	6	4.8
	60th	73	58.4
	70th	42	33.6
	80th	4	3.2
Patients	CT	54	43.2
	MRI	71	56.8
Radiological T	T1	46	36.8
	T2	46	36.8
	T3	22	17.6
	T4	11	8.8
Histopathological T	T1	31	24.8
	T2	49	39.2
	T3	28	22.4
	T4	17	13.6
Radiological DOI	≤10 mm	47	37.6
	>10 mm	78	62.4
Histopathological DOI	≤10 mm	69	55.2
	>10 mm	56	44.8
Radiological nodal examination	No	74	59.2
	Yes	51	40.8
Histopathological nodal examination	No	89	71.2
	Yes	36	28.8

**Table 2 diagnostics-13-03578-t002:** Summary of characteristic values (* statistical significance; *p*-value ≤ 0.05).

		All	MRI	*p*-Value	CT	*p*-Value
T T1/T2/T3/T4	Histopathological	31/49/28/17	23/33/12/3	NA	8/16/16/14	0.07
	Radiological	46/46/22/11	32/24/12/3		14/22/10/8	
DOI (≤10 mm, >10 mm)	Histopathological	69/56	46/25	0.006 *	23/31	0.23
	Radiological	47/78	29/42		18/36	
Nodal metastasis (N0/N+)	Histopathological	87/38	57/14	0.10	30/24	0.02 *
	Radiological	70/54	49/22		21/32	

NA—not available.

## Data Availability

The data on which this study is based will be made available upon request.

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
