# Peer review of "Efficacy of Computed Tomography and Magnetic Resonance Imaging in the Assessment of Depth of Invasion in Oral Squamous Cell Carcinoma: Analysis of 125 Clinical Cases"

_diagnostics, 2023, doi:10.3390/diagnostics13233578_

Round 1

Reviewer 1 Report

Comments and Suggestions for Authors

The authors described "Efficacy of CT and MRI in assessment of DOI in oral squamous cell carcinoma. Analysis of 125 clinical cases." in a retrospective study. This topic should be informative and attractive for head and neck surgeons. However, there are a lot of studies related to this topic, and conclusions are similar to this manuscript. The presented study emphasized how difficult preoperative radiological evaluation of OSCC can be, despite the increasing accuracy of CT and MRI. Both studies have shown relatively high imaging efficacy, but it is impossible to identify a study that is superior in all clinical cases. CT and MRI imaging efficacy has been proven to decrease with tumor growth (T and DOI). This result should be especially taken into account when evaluating large tumors at the border of operability. These are already proven. There were no what's new in this manuscript. So, I have some suggestions and the authors should add them.

1. As they mentioned, an examination that can be used in depicting not only the primary focus but also the lymphatic system is PET-CT. Why did not you examine PET-CT? Please add the reason.

2. Head and neck surgeons have an interest in sentinel lymph nodes metastasis. How about the evaluation of one or two lymph nodes swelling?  Please focus on the sentinel lymph node metastasis, add the analysis of pre-and post-surgical histopathological results.

3. IRB number should be added. Because the authors performed statistical analysis.

Comments on the Quality of English Language

I recommend English editing.

Author Response

Dear Reviewer

First of all, I would like to thank You for the review and such valuable comments. They will certainly contribute to increasing the value of the manuscript.

Extensive English editing was done by MDPI using an experienced, native English-speaking editor.

 According to Your questions:

  1. I agree that PET-CT is a test with high sensitivity and specificity and also allows for the assessment of the lymphatic system of the neck. However, at our institution, the standard protocol for head and neck cancer patients includes CT or MRI of the head and neck and CT of the chest, and abdominal ultrasound. It does not include PET-CT imaging. Of course, in cases of ambiguous test results in the basic protocol (M feature), PET-CT was performed. However, due to the qualification criteria of the study group, patients with the M+ trait were not qualified. This is because the M+ feature is most often disqualified from surgical treatment (except for single cases of surgery of the primary tumor, the lymphatic system of the neck, and metastasectomy). Thank You for pointing this out, this information will be included in the manuscript
  2. Thank you for this question. Indeed, sentinel lymph node mapping is one of the methods for imaging the lymphatic system of the neck in patients in the initial phase of OSCC. Sentinel lymph node biopsy (SLNB) is also not a standard operating procedure at our institution. SLNB is, of course, often performed for other anatomical areas, but due to the diffuse lymphatic system of the neck, it is not routinely used in oral cancers.

In the available literature, no clear results are confirming the benefits of using SLNB over the use of elective neck dissection. Some authors also try to compare the depth of DOI with SLNB, finding the cut-off point seems very difficult. Knowing the relationship between the increasing risk of metastases and increasing DOI, it seems unreasonable to perform SLNB among patients with large DOI (>10 mm). In our study, 62.4% of patients had DOI > 10 mm. Of course, I agree with Your opinion that the assessment of sentinel lymph nodes is an extremely interesting element of the assessment of nodal metastasis, but due to the fact that the analysis was performed retrospectively, the analysis of sentinel lymph node metastasis in this group is not possible. Thank You again for pointing this out. I hope that in future publications related to OSCC, I will be able to include research on the sentinel lymph node.

  1. Thank you for pointing this out. Of course, I agree with Your opinion. The Institutional Review Board Statement was sent to the editorial office the following day after submission. The lack of IRB approval was an oversight after many hours of work on the manuscript on the day of submission. Sorry for this error. The IRB number has already been completed in the manuscript.

I would like to thank You again for your positive review. Of course, I agree that in the available literature, we can find numerous articles related to OSCC imaging. However, I would like to point out that in our manuscript we performed a multivariate analysis on a quite large group of patients (125) along with a detailed statistical analysis. The discussion also contains the results of studies available in the literature and, as it was shown, their results differ from each other. This justifies further research into the diagnosis of OSCC.

Yours faithfully

Reviewer 2 Report

Comments and Suggestions for Authors

The topic addressed in the article is interesting to contribute to clinical decisions regarding the treatment of OSCC. Furthermore, it is well structured and the methodology is appropriate. However, I suggest including in the introduction more data from other studies on the accuracy of using images for size and DOI determination. Furthermore, I suggest that the criteria used to classify the radiological T (T1-T2-T3-T4) be explained in detail.

Finally, I suggest that for a better understanding of the results, the methodological difference between the calculation of the histopatological DOI and the radiological DOI be discussed.

Author Response

Dear Reviewer.

First of all, I would like to thank You for the review and all the comments. Their addition to the manuscript will certainly increase its scientific value and contribute to better exploration of the topic. Referring directly to Your questions, the introduction has been supplemented with methods used for preoperative assessment of DOI. It also describes the reason why the remaining methods were not included in our study.

The tumor size was assessed in two orthogonal axes according to the 8th edition AJCC classification.

“Finally, I suggest that for a better understanding of the results, the methodological difference between the calculation of the histopatological DOI and the radiological DOI be discussed.”

 Indeed, this is an extremely important topic that is not sufficiently explained in the text. Thank You for pointing this out. The manuscript has been updated with this important information. As is known, in some cases, DOI assessment according to the 8th edition AJCC can be difficult. For this reason, in order to be able to compare radiological results with histopathological DOI, we used criteria excluding exophytic and deeply ulcerated tumors. In these cases, it is clear that the results of pre- and post-treatment measurements will differ significantly. The use of such criteria allowed us to make the results credible.

I hope that the current version of the manuscript will meet Your expectations.

Yours faithfully

Reviewer 3 Report

Comments and Suggestions for Authors

The study is very interesting for the surgeon in order to decide the best treatment choice for the neck. However it not clear how it is comparable a pathological finding with clinical or radiological data. Depth of invasion is measured from the basement membrane of the epithelium from which tumor arise to deepest point of invasion.

Please, the authors explain this point

Author Response

Dear Reviewer.

First of all, I would like to thank You for the review. Of course, Your comments will be taken into account in the manuscript, which will certainly increase its scientific value. Indeed, the DOI assessment was not adequately described. This is very important, so thank You again for noticing this problem. In many publications, there is no distinction between tumor thickness (TT) and DOI which is obviously not the correct procedure. Referring to the research results of other authors, among others (e.g. Abeer M. Salama; doi:10.1111/his.14291), we still find many technical and methodological problems in assessing DOI according to the 8th Edition AJCC guidelines, including in the case of polypoid (exophytic) tumors and those with deep ulceration. For this reason, to be able to compare the radiological and histopathological DOI results, patients with deeply ulcerated tumors and exophytic lesions were excluded from the study. We deliberately did not introduce the concept of tumor thickness TT, which, according to the 8th Edition of the AJCC, is not included in the clinical classification. In the case of other tumors (without deep ulceration or exophytic growth), the difference between radiological DOI and histopathological DOI resulting from the measurement methodology is very small (0.5 - 1 mm). This allowed for a reliable assessment of the effectiveness of radiological imaging in relation to histopathological results.

Thank You again for drawing attention to this important issue

Yours faithfully

Round 2

Reviewer 1 Report

Comments and Suggestions for Authors

The authors revised the manuscript precisely. However, I am wondering whether this study has a novelty or not. I leave the decision to the editor.